# Successful expansion and cryopreservation of human natural killer cell line NK-92 for clinical manufacturing

Seul Lee[1,2], Yunjoo Joo[1,2], Eun Ji Lee[1,2], Youngseon Byeon[1], Jae-Hwan Kim[1], Kyoung-Ho Pyo[1,3], Young Seob Kim[1], Sun Min Lim[3], Peter Kilbride[4], Rohin K. Iyer[5], Mingming Li[6], Mandy C. French[7], Jung-Yub Lee[8], Jeeheon Kang[8], Hyesin Byun[8], Byoung Chul Cho[3]*

1 Severance Biomedical Science Institute, Yonsei University College of Medicine, Seoul, Korea, 2 Brain Korea 21 PLUS Project for Medical Science, Yonsei University College of Medicine, Seoul, Korea, 3 Division of Medical Oncology, Yonsei Cancer Center, Yonsei University College of Medicine, Seoul, Korea, 4 Global Life Sciences Solutions, Cambridge, United Kingdom, 5 Global Life Sciences Solutions USA LLC 100 Results Way, Marlborough, MA, United States of America, 6 Global Life Sciences Solutions Singapore Pte. Ltd., HarbourFront Center, Singapore, Singapore, 7 Global Life Sciences Technologies (Shanghai) Co., Ltd., Shanghai Municipality, Shanghai, China, 8 Global Life Sciences Solutions Korea Limited 5F, Gangnam-gu, Seoul, Korea

* cbc1971@yuhs.ac

**Data Availability Statement:** All relevant data are within the paper and its Supporting Information files.

## Abstract

Natural killer (NK) cells have recently shown renewed promise as therapeutic cells for use in treating hematologic cancer indications. Despite this promise, NK cell manufacturing workflows remain largely manual, open, and disconnected, and depend on feeders, as well as outdated unit operations or processes, often utilizing research-grade reagents. Successful scale-up of NK cells critically depends on the availability and performance of nutrient-rich expansion media and cryopreservation conditions that are conducive to high cell viability and recovery post-thaw. In this paper we used Cytiva hardware and media to expand the NK92 cell line in a model process that is suitable for GMP and clinical manufacturing of NK cells. We tested a range of cryopreservation factors including cooling rate, a range of DMSO-containing and DMSO-free cryoprotectants, ice nucleation, and cell density. Higher post-thaw recovery was seen in cryobags over cryovials cooled in identical conditions, and cooling rates of 1°C/min or 2°C/min optimal for cryopreservation in DMSO-containing and DMSO-free cryoprotectants respectively. Higher cell densities of $5x10^7$ cells/ml gave higher post-thaw viability than those cryopreserved at either $1x10^6$ or $5x10^6$ cells/ml. This enabled us to automate, close and connect unit operations within the workflow while demonstrating superior expansion and cryopreservation of NK92 cells. Cellular outputs and performance were conducive to clinical dosing regimens, serving as a proof-of-concept for future clinical and commercial manufacturing.

## Introduction

Axicabtagene ciloleucel [1] and Tisagenlecleucel [2], developed by Gilead and Novartis, respectively, are two cellular immunotherapies that are already available on market and used

**Funding:** National Research Foundation of Korea (NRF) through a grant from the Korean government (MSIT) under Grant No. 2022R1A2C3005817.

**Competing interests:** NO authors have competing interests

in cancer treatment for B-cell malignancies, providing better patient recovery for currently incurable diseases. With these successes clinically, more cell therapy products are being developed for various types of cancers and many of them have entered into phase 2/3 clinical trials [3]. Geographically, more than 50% of the total cell therapy development is being conducted in North America (USA) (344 cases as of 2018), and a significant portion of the research is being conducted in China (203 cases) [4].

Natural Killer (NK) cells are cytotoxic lymphocytes and a significant constituent of the innate immune system. NK cells do not present the TCR (T cell receptor) complex and common T cell surface receptors such as CD4 or immunoglobulin, so they are unique in comparison to conventional T cells and B cells in this regard [5]. NK cells, which generally represent up to 5–15% of circulating lymphocytes (and this can be lower in cancer patients), do not recognize target cells through specific antigen binding; in contrast, they are able to target and bind to many malignant cells, including virus-infected cells, without prior immune sensitization [6]. Through this mechanism, targeted cells are eliminated through cell lysis.

NK cells can kill cancer cells non-specifically; activated NK cells directly lyse tumor cells by releasing cytotoxic granules containing perforin and granzyme, similar to activated cytotoxic T cells [7]. NK cells are also potent producers of chemokines and cytokines such as interferon-gamma (IFN-γ) and tumor necrosis factor-alpha (TNF-α), thus essential for regulating the adaptive immune response.

Therapeutic NK cells can be isolated from the peripheral blood lymphocytes of a patient, and then reinfused to the same donor or other patients after cell expansion. This modality of isolating therapeutic NK cells, followed by *ex vivo* expansion and subsequent *in vivo* infusion therapy, can induce highly effective apoptosis in virally infected or tumorigenic cells. To effectively use NK cells for immune cell therapy, especially for allogeneic cancer therapies, a large number of NK cells is typically required (e.g. billions of cells per dose), motivating the use of scalable expansion and cryopreservation solutions.

Unfortunately, NK cells expanded using legacy technologies exhibit inconsistency in proliferation and cytotoxic activity. In addition, donor-to-donor variation and cellular heterogeneity creates difficulties with standardization of NK cell products. To overcome these limitations, researchers have attempted to use stable NK cell lines for clinical applications.

The NK-92 cell line is a human IL-2-dependent NK cell line established from the peripheral blood mononuclear cells (PBMC) of a 50-year-old male diagnosed with non-Hodgkin's lymphoma [8]. NK-92 cells are characterized by CD56 bright and CD2+, in the absence of CD3, CD8, and CD16. Additionally, NK-92 cells express high levels of perforin and granzyme B. They have a comprehensive cytotoxic range and are active against hematological malignancies and cell lines derived from solid tumors with persistent and high cytotoxicity against cancer targets. NK-92 cells have undergone extensive preclinical development and have completed phase I trials in cancer patients [9–11]. The NK-92 line has also been infused into end-stage cancer patients, demonstrating clinical benefit with few adverse events [12]. Expanded *ex vivo* under IL-2-supplemented conditions, the NK-92 cell line comprises a homogeneous cell population that can be cryopreserved for long term storage [12].

Primary NK cells are traditionally difficult to cryopreserve, however, and few studies have evaluated the effects of cryopreservation on NK cell expansion and function [13–15]. As such, overcoming technical limitations with cryopreservation of NK cells remains a challenge. A (good manufacturing process) GMP-compliant, closed and automated culture method for clinical application is also needed to ensure reproducibility and scalability [16, 17]. The availability of well-established cell lines like the NK-92 line, together with better automation and standardization of expansion and cryopreservation parameters, offers hope in overcoming these challenges. Current methods in cell therapies more generally use around 1°C/min

cooling rates, with Dimethyl sulfoxide (DMSO) and DMSO-free cryoprotectants common [9, 18]–no common cryopreservation strategy exists for these cells between different groups.

Cryopreservation of an NK92 working cell bank can help surmount difficulties associated with maintaining a continuously grown fresh cell culture, immediately fulfilling the need of dosing requirements at any time for a patient, and thus represents an important technical challenge to overcome [13]. Cryopreservation allows for economies of scale to be introduced, reducing costs of treatment, and allows for flexibility in patient scheduling [16, 17].

In addition, the risk of infusing a unit contaminated with bacteria or mycoplasma can be prevented through rigorous sterility testing prior to cryopreserving the cells, however this possibility is more difficult with fresh cell therapy preparations due to the lengthy time required to carry out these tests. Moreover, the practical challenges of working on primary cells from various donors or batches is eliminated by using a homogeneous, off-the-shelf product, along with the risk of variability and cross-contamination between donors in a manufacturing setting [16, 17].

Cryopreservation requires cells to be protected from damage caused by ice, high solute concentrations on cooling, and direct cold damage. This is achieved through the use of cryoprotectants (CPAs) [19]. DMSO-based CPAs are the most commonly used in the cryopreservation of cell therapies, though concerns have been raised around the toxic effect of this agent and DMSO-free CPAs are under active consideration [13, 20–22]. The choice of CPA must be taken carefully for cell therapies–not only must they optimize biological outcome and function, but must exclude sources of variability or potential contamination (for example, as may occur with commonly used animal-derived serums [16, 17] and human albumins [15, 22]).

Addition of the CPA is followed by controlled rate freezing prior to long term storage in liquid nitrogen vapor at or below -120˚C. It has long been known that cooling rate must be carefully controlled to ensure good post-thaw viability, with each cell type having its own specific optimal rate, with controlled rate freezers being the most common way to achieve this [23, 24]. Practically speaking, cryopreservation must offer stable long-term storage for a sufficient quantity of cells to be applicable to GMP cell therapy manufacturing, so the impact of extended low temperature storage time and cell concentration must be understood.

In this study, we propose an effective method for the expansion and cryopreservation of NK-92 cells for clinical research and industrialization. Using commercially available hardware (Cytiva), media and cell freezing solutions that are currently marketed as GMP grade, we demonstrate a scalable workflow for the expansion and cryopreservation of NK-92 cells. In addition to automating and functionally closing the process, we also verified excellent culture kinetics of genetically engineered NK-92 cells.

## Materials and methods

### Cell line

The human NK cell line NK-92 was purchased from American Type Culture Collection (ATCC, CRL-2407). Fc receptor (CD16) expressed NK-92 (CD16-NK-92) was purchased from American Type Culture Collection (ATCC, PTA-8836). NK-92 expressing T cell receptor (TCR-NK-92) was genetically engineered using lentivirus (LVV) in previous studies.

### Culture media

To identify the optimal media to expand NK-92 cells, three medium formulations were assessed for their ability to support NK-92 expansion and maintain viability, namely:

- X-VIVO 10 (Lonza)

- Xuri T cell expansion medium (Cytiva)

- NK MACS (Miltenyi)

Cryopreserved cells were thawed rapidly in a water bath at 37°C and immediately cultured in each medium formulation. Cell proliferation and viability were assessed over a defined culture period to identify media formulations that could support NK-92 growth. Cells were plated in triplicates in a T75 flask at a concentration of 2.5 x $10^5$ cells/mL. Cell expansion (assessed by haemocytometer), viability (Trypan blue exclusion method or apoptosis assay), and proliferation (Ki67 staining) were determined over a 12-day period.

## Cryopreservation

Three solution formulations were assessed for their ability to support NK-92 viability and expansion, namely: CryoSOfree DMSO-free (Sigma), CryoStor CS10 (STEMCELL technologies, containing 10% DMSO) and Stem-Cell banker DMSO Free (AMSBIO). Freezing was performed using the VIA Freeze (Cytiva) liquid nitrogen-free controlled rate freezer (CRF). Cryopreservation conditions were optimized under the following conditions after the cells were loaded into the VIA Freeze at 4°C. The chamber temperature was set to decrease at -0.5°C/min, -1°C/min, or -2°C/min after 10 min holding at 4°C for stabilization. Upon reaching -80°C, samples were held for a minimum of 10 min for temperature stabilization before transferring to liquid nitrogen (vapor phase) storage tanks.

Efficiency of cryopreservation efficiency according to induced ice nucleation was also determined following the above protocol with -10°C hold for 7 min (temperature stabilization). Vials were then taken out to mix them by inversion (to establish a uniform cell concentration throughout each vial), and ice nucleation was manually induced by tapping each vial after inversion followed by a further hold of 15 min for temperature stabilization. Efficiency of cryopreservation as a function of cell concentration was also studied with 5.0 x $10^6$ cells/mL/vial, 1.0 x $10^7$ cells/mL/vial, and 5.0 x $10^7$ cells/mL/vial. For each condition described above, experiments were repeated five times. Cell viability was measured using Trypan blue. Cell viability was measured 24 hour and 72 hour after thawing in a 37°C water bath.

## NK-92 cells expansion

**Culture using manual method in T flask.** Cells were initially thawed in culture media supplemented with 5% heat-inactivated human AB serum (Sigma) and 250 U/mL of Proleukin [2]. NK-92 cell concentration was controlled under standard conditions between 0.25–1.0 x $10^6$ cells/mL and media were replaced once every 2–3 days for optimized proliferation. To prevent inhibition of cell growth due to cell-cell contact, the cells were split into multiple flasks when necessary. All cell culture was conducted in a BSL-2 environment under strict antibiotic-free conditions.

**Culture in closed expansion bioreactor using Xuri W25 (Cytiva).** The conditions for Xuri W25 (Cytiva) rocking bioreactor were controlled at all times with setpoints of 37°C temperature, 5% $CO_2$, pH between 7.2~7.4, airflow of 0.1l/min, rocking rate 6 RPM, and rocking angle at 2°. Once a minimal number of $1 \times 10^8$ cells was obtained in manual culture in T75 flask, cells were transferred to two 2L Xuri Cellbag bioreactors with perfusion filter (Cytiva). In parallel, once a minimal number of $2.5 \times 10^8$ cells were obtained in manual culture in a T75 flask, cells were transferred to 10L Xuri Cellbags with perfusion filter.

After cell inoculation to the Xuri Cellbag bioreactor, the medium was added until reaching a maximum of 1L (2L bag) or 5L (10L bag) depending on the bag capacity, while maintaining the cells at $2.5 \times 10^5$ cells per mL.

In the culture propagation phase before reaching the maximum volume, when the cell concentration reached $0.5 \sim 1.0 \times 10^6$ cells/mL, half of the culture media was removed using a perfusion line. And fresh media was replenished as needed to maintain $2.5 \times 10^5$ cells per mL.

In **Method 1** used continuous perfusion function in Xuri system.

After reaching the maximum volume, the media was changed using continuous perfusion 50% rate of total volume per day.

In **Method 2** used not continuous perfusion function in Xuri system.

After reaching the maximum volume, half of the entire media was completely removed and then culturing was performed by adding a new media.

**Apoptosis and proliferation analysis.** For identification of pre- and pro-apoptotic and necrotic cells in culture, NK-92 cells were stained with Annexin V and propidium iodide according to manufacturer's protocol (FITC Annexin V Apoptosis Detection Kit I, BD Biosciences). To assess target cell killing efficacy, one batch of effector cells was thawed, and they were seeded at E:T ratios of 0.5:1, 1:1. K562 cells were stained with 7AAD (BD Biosciences). Frozen cells were thawed and analyzed immediately. For proliferation analysis, 70% ethanol was prepared and chilled to -20°C. Target cells of interest were prepared and washed 2X with PBS, centrifuging at 400 xg for 5 minutes. The supernatant was discarded and the cell pellet loosened by vortexing. 3ml of chilled 70% ethanol was added drop by drop to the cell pellet while vortexing. Vortexing continued for 30 seconds and then incubated at -20°C for 1 hour and then stained with KI-67.

**NK-92 cell analysis by flow cytometry.** Cells were harvested and washed extensively, and biomarkers were assessed by flow cytometric analysis (see below) using specific antibodies. Antibodies (Biolegend) specific for NKG2D, NKp30, NKp44, and NKp46 were used to profile the activation receptors; For the analysis of effector molecule expression, one batch of effector cells was thawed and subsequently seeded at E:T ratios of 0.5:1 and 1:1. The expression of CD107a, DNAM-1, Granzyme B, TNF-α, IFN-γ, and Perforin was then assessed to profile the effector molecules.; KIR2D, TIGIT, TIM3, and PD-1 to profile the inhibitory receptors. All antibody staining processes for flow cytometry were done according to the protocol below.

For surface antigen staining, cells were washed once with PBS (containing 2% FBS and 1 mM EDTA) and incubated with appropriate amounts of antibody at 4°C for 30 min. The labeled cells were then washed with PBS and data acquisition was performed. For intracellular staining, cells were fixed and permeabilized (eBioscience™ Fixation/Permeabilization Diluent, Invitrogen, #00-5223-56) for 15 min in a solution containing 1% PFA in 1× permeabilization wash buffer (Permeabilization buffer 10X, Invitrogen, Cat# 00-8333-56). After two washes with permeabilization wash buffer, the cells were incubated with appropriate amounts of antibody at 4°C for 30 min. The labeled cells were then washed with permeabilization wash buffer and data acquisition was carried out. Cells were acquired on LSR Fortessa (BD Biosciences) and data were analyzed using FlowJo software (Treestar Inc.)

## Statistics

Data are reported as the mean ± standard error of the mean [25]. All data was performed using analysis of variance [26] with the Mantel-Cox log-rank test of significant difference in GraphPad Prism (GraphPad Prism version 7.00 for Windows; GraphPad Software). Statistical analysis of flow cytometry was performed using a t-test in GraphPad Prism.

## Results

### Assessment of cryopreservation of NK-92 cells

- *Cooling rate and cryoprotectant*

Cell viability was evaluated as a function of cooling rate and cryoprotectant. **Fig 1A** is temperature decrease of -0.5˚C/min, **Fig 1B** -1.0˚C/min and **Fig 1C**, -2.0˚C/min.

All three types of CPA showed the best survival rate following a cooling rate of -1.0˚C/min. In particular, cells frozen in STEM CELL BANKER DMSO-free CPA showed a tendency to both survive and recover post-thaw, showing that cells maintained proliferation ability after thawing under condition of cooling rate of -1.0˚C/min.

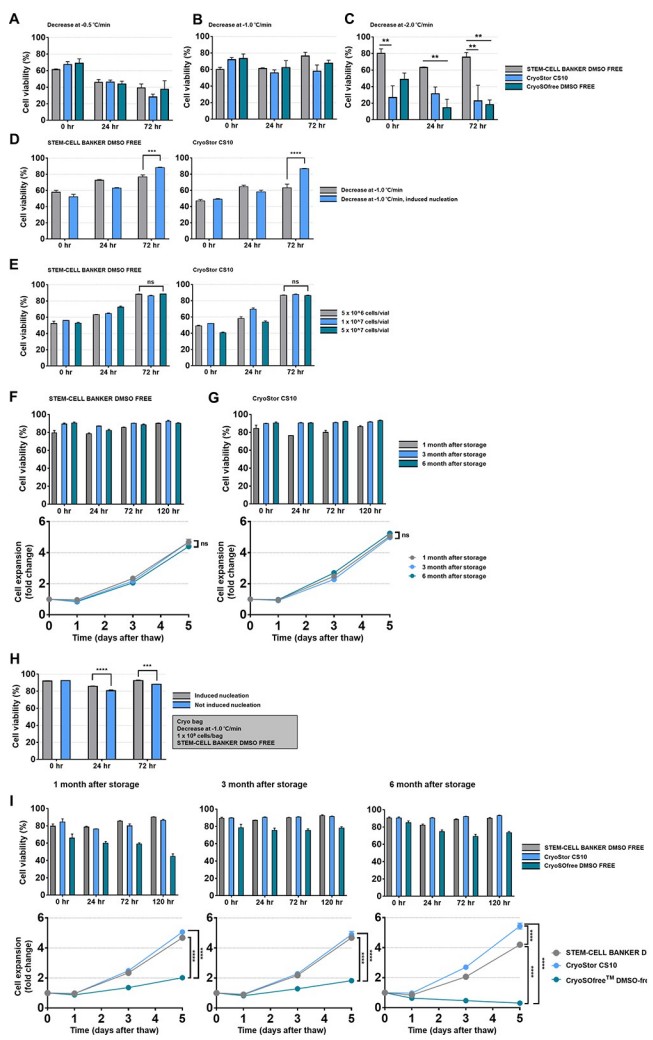

**Figure 1. C**

| Sidak's multiple comparisons test | Mean Diff. | 95.00% CI of diff. | Significant? | Summary | Adjusted P Value |
|---|---|---|---|---|---|
| **0 hr** | | | | | |
| STEM-CELL BANKER DMSO FREE vs. CryoStor CS10 | 53.44 | 16.44 to 90.44 | Yes | ** | 0.0039 |
| **24 hr** | | | | | |
| STEM-CELL BANKER DMSO FREE vs. CryoSOfree DMSO-free | 48.63 | 11.62 to 85.63 | Yes | ** | 0.0084 |
| **72 hr** | | | | | |
| STEM-CELL BANKER DMSO FREE vs. CryoStor CS10 | 52.74 | 15.73 to 89.74 | Yes | ** | 0.0044 |
| STEM-CELL BANKER DMSO FREE vs. CryoSOfree DMSO-free | 57.37 | 20.36 to 94.37 | Yes | ** | 0.0021 |

**Figure 1. D**

| Sidak's multiple comparisons test | Mean Diff. | 95.00% CI of diff. | Significant? | Summary | Adjusted P Value |
|---|---|---|---|---|---|
| **72 hr** | | | | | |
| STEM-CELL BANKER DMSO FREE | -11.52 | -18.58 to -4.467 | Yes | *** | 0.0010 |
| CryoStor CS10 | -23.58 | -31.95 to -15.21 | Yes | **** | <0.0001 |

**Figure 1. H**

| Sidak's multiple comparisons test | Mean Diff. | 95.00% CI of diff. | Significant? | Summary | Adjusted P Value |
|---|---|---|---|---|---|
| **0 hr** | | | | | |
| with Nucleation vs without Nucleation | -0.41 | -2.794 to 1.974 | Yes | ns | 0.9617 |
| **24 hr** | | | | | |
| with Nucleation vs without Nucleation | 5.152 | 2.768 to 7.536 | Yes | **** | <0.0001 |
| **72 hr** | | | | | |
| with Nucleation vs without Nucleation | 4.306 | 1.922 to 6.69 | Yes | *** | 0.003 |

**Fig 1. Assessment cell viability after cryopreservation for NK-92.** Cell viability was evaluated according to the freezing rate. Cell viability after thawing by type of cryoprotective agents (A) frozen at a rate of -0.5˚C/min, (B) a rate of -1.0˚C/min, and (C) a rate of -2.0˚C/min. Cell viability was evaluated according to nucleation induction during cell freezing and viability results according to cell concentration. (D) Cell viability after thawing as a function of cell concentration, (E) Cell viability after thawing according to cell concentration induced nucleation. Cell viability was assessed after long-term storage of frozen cells. Cell viability and expansion efficacy of (F) 1 month, (G) 3 months, and (H) 6 months after storage of frozen cells in liquid nitrogen vapor phase. Evaluation of preservation efficacy in cryo-bags (I)Assessment cell viability after cryopreservation for NK-92. Cell viability was assessed after long-term storage of frozen cells. Cell viability and expansion efficacy of 1 month, 3 months, and 6 months after storage frozen cells. p value: ns > 0.05, ** < 0.01, *** < 0.001, **** < 0.0001.

There was no statistically significant difference in cell viability after thawing of the three types of CPA frozen at -0.5°C/min and -1.0°C/min. Interestingly, in the condition frozen at -2.0°C/min, when freezing with STEM-CELL BANKER DMSO FREE, there was a statistically significant difference compared with the other two CPAs.

In addition, cell viability was evaluated according to nucleation induction during cell freezing and viability results according to cell concentration.

- *Ice nucleation*

To evaluate the impact of ice nucleation to viability efficiency, cell viability was evaluated after thawing by inducing nucleation under the optimal conditions from above **Fig 1B** (decrease of -1.0 C/min). These experiments were conducted using two CPAs that are best efficient in previous experiments (STEM-CELL BANKER DMSO FREE and CryoStor CS10).

There was no statistically significant difference in cell viability at 0 hour and 24hour after thawing of the two types of CPA frozen. Interestingly, at 72 hours after thawing, when both CPAs induced nucleation, a significant increase in cell viability was observed (**Fig 1D**).

Inducing nucleation during cell freezing can aid in cell recovery after thawing.

- *Cell concentration*

We used the optimized freezing conditions previously identified, performed cell freezing cryopreservation at three cell densities: $5 \times 10^6$ cells/mL, $1 \times 10^7$ cells/mL, and $5 \times 10^7$ cells/mL, as shown in **Fig 1E**. No significant difference was observed between any measured cell concentration.

- *Storage time*

The impact of storage time in LN2 vapour phase is shown in **Fig 1F–1G** (using the optimized protocol from **Fig 1A–1E**–a decrease of -1.0°C/min, $5 \times 10^7$ cells/vial, induced nucleation). Cryopreserved cells were stored in a nitrogen tank for 1, 3, and 6 months, and viability and proliferation efficiency were evaluated after thawing.

In both CPAs, no significant difference was seen between short-term and long-term storage in terms of cellular viability and cell proliferation.

This suggests that the optimized freezing conditions are suitable for long-term cryo-preservation of NK-92 cells.

However, the proliferation of the CryoSOFree was significantly reduced after 6 months of storage compared to 1 or 3 months of storage.

Finally, the optimized parameters were used to scale cryopreservation from cryovials to cryobags. Cells were cryopreserved at a concentration of $5 \times 10^7$ cells/mL and at a cooling rate of -1.0°C/min using STEM CELL BANKER DMSO-free cryoprotectant. It was confirmed that the survival rate of the nucleation-induced group under the above conditions was improved by 5–10% compared to the group's survival rate that did not induce nucleation.

In particular, in the case of the cell freezing bag, unlike the vial for cell bank production, has less sample to sample variation during cryopreservation, and the survival rate is 80–90% or more immediately after thawing (**Fig 1H** compared with cryovials in **Fig 1B**). As confirmed from **S1 Fig**, cell aggregation was observed in all cells cultured in X-VIVO 10 media after thawing. In contrast, cell aggregation was not observed in all conditions thawed in Xuri media. Therefore, the best survival rate for NK92 cells was observed in the condition frozen using STEM CELL BANKER-DMSO free or CryoStor CS10, under a decrease of -1.0°C/min and thawed with Xuri T cell expansion media.

## Assessment of cell stability after thawing (NK-92)

We compared the proliferation efficiency for NK-92 cells by using GMP grade culture media for immune cells commonly used for cell therapy. NK-92 cells were thawed and then cultured in each medium to compare the results. NK-92 cells were purchased through ATCC and frozen at $5 \times 10^7$ cells/mL using CryoStor® CS10 agent recommended by ATCC. After thawing, the NK-92 was diluted to $2.5 \times 10^5$ cells/mL in each medium, and culture expansion was started.

In **Fig 2A**, using Xuri T-cell expansion media, the cells proliferated 2.1-fold on the 3rd day, 4.4-fold on the 6th day, and 11.2-fold on the 9th day after thawing. Using X-VIVO 10 media, cell number increased 1.24-fold on the 3rd day, 1.84-fold on the 6th day, and 4.53-fold on the 9th day after thawing. On the other hand, in the case of NK MACS media, cell proliferation was not observed after thawing, and the number of cells decreased during the post-thaw culture period.

This observation was also confirmed by checking the morphology of NK-92 cells after thawing. A significant number of floating cells could be seen that formed colonies in culture in the case of Xuri T-cell medium, which is a normal sign of proliferation, however in the case of NK MACS media or X-VIVO 10 media, we noticed that colony formation was not occurring as readily, as outlined below.

As shown in **Fig 2B**, normal colony formation is formed in Xuri T-cell expansion media after thawing. The culture was stabilized one day after thawing, followed by active colony formation after three days. On the other hand, NK-92 cells thawed in X-VIVO 10 and NK MACS media were not stabilized, and instead exhibited cell clumps after thawing. It was confirmed that the aggregated cells were mostly dead cells when stained with Trypan blue (not shown). X-VIVO 10 media was able to continuously support the release of living cells from the aggregated cells into suspension, and it was observed that these cells proliferated, but did so more gradually. NK-92 expansion in NK MACS media, in contrast, did not induce cell proliferation, with a subsequent decline in cell number and viability three days after thawing.

It is well known that NK-92 cells are IL-2-dependent, highly susceptible to cell culture conditions, as well as highly dependent on the success of freezing and thawing for viability and function [27, 28]. Some of the cell death observed upon thaw was likely induced by freezing and thawing, and it was confirmed that it was affected by the choice of media used post-thaw as well. Dead cells were not observed in the Xuri T-cell expansion medium, which is presumed to be due to the composition of the media, including carbon sources (such as sugars and amino acids) and other proteins present to support cell maintenance and growth. Inhibition of cell aggregation was also observed in Xuri T-cell expansion media, which is presumed to have enhanced cell viability by preventing necrosis resulting from the release of lytic granules such as Granzyme, often secreted during aggregation.

Furthermore, it is essential to assess the post-thaw cytotoxicity of NK-92 cells. We thawed NK-92 cells using Xuri T-cell expansion medium and allowed them to stabilize for 5 days. Subsequently, we initiated co-cultures of NK-92 cells with K562 cells at effector-to-target (E:T) ratios of 0.5:1, 1:1, and 2:1 for a 24-hour duration.

We conducted flow cytometry analysis on a subset of 10,000 cells from a batch. While statistical significance was not achieved due to the limited sample size, we observed a consistent trend indicating the stability of both the target killing effect and the expression of effector molecules, even after the thawing of NK92 cells.

In **Fig 2C**, we conducted an analysis of cytotoxicity by evaluating the proportion of 7AAD + cells among the K562 cell population. The results revealed a 60% increase at the 0.5:1 E:T ratio and an 80% increase at the highest E:T ratio.

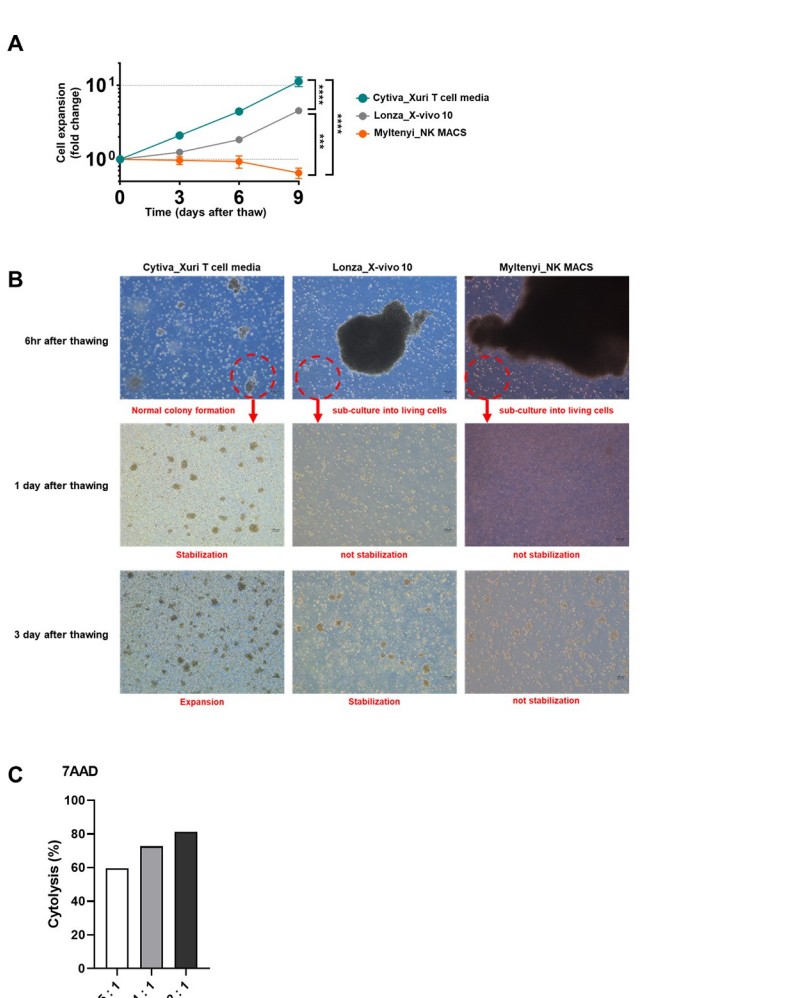

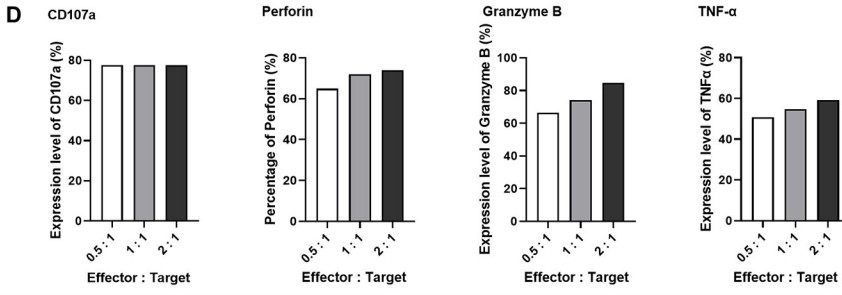

**Fig 2. Assessment of cell stability after thawing.** Cell stability using three types of cell culture media. (A) Cell expansion efficacy of the NK-92 cell line. (B) Cell morphology as a function of days in culture post-thaw. (C) The efficacy of NK-92 in target cell killing. (D) The expression level of effector molecules in co-culture. Error bars represent standard error of the mean [25]. p value: *** < 0.001, **** < 0.0001.

In **Fig 2D**, we illustrated the expression of effector molecules in NK-92 cells at the 24-hour mark. CD107a showed a consistent range of 77–78%, which did not appear to be dependent on the E:T ratio. In contrast, the expression levels of Perforin, Granzyme B, and TNF-α seemed to be influenced by the E:T ratio, with higher levels observed as the ratio increased. Notably, these molecules were present in significant concentrations even at the 0.5:1 ratio.

Additionally, the overall cytotoxicity of NK-92 cells was 20–35% at 4 hours and 40–80% at 16 hours in a 1:1 E:T ratio. The expression level of CD107a+ NK-92 cells was 40–60% at 5 hours [29].

## Evaluation of cell expansion efficiency according to the culture medium (NK-92)

In **Fig 3**, Expansion efficiency was assessed for comparison of the two media. For NK-92 cells, cell number and viability, and proliferation rate were checked every two days, and the expression level of cell activator and effector receptors was confirmed after culture up to 12 days. NK-92 cells were observed for each factor using stabilized cells after thawing.

In **Fig 3A–3D**, the cell concentration, expansion efficiency, viability, and proliferation rate of the proliferated NK-92 cells were checked every two days. It was confirmed that Xuri T-cell expansion medium and X-VIVO 10 medium had no significant functional difference in the expansion of NK-92 cells.

NK-92 cells were assessed for changes in expression of activation receptors and effector molecules due to the difference in the two media components. To establish the cause of damage, 12 surface factors were studied. The expression level of each factor was analyzed through FACS analysis. In **Fig 3E**, cell surface factors NKG2D, NKp30, NKp44, and NKp46 were observed, and in **Fig 3F**, CD107a, Granzyme B, and Perforin were identified as effector molecules. This gives us a greater insight to cryopreservation damage over other tests such as cytotoxicity to a target cell line, which may only give a general output dependent on target cell line used as opposed to specific cryopreservation damage to the cells. There were differences in the expression levels of these receptors in Xuri T-cell expansion medium vs. X-VIVO 10 medium for each factor, but not significant at $p < 0.05$.

In **S2 Fig**, differences in cytokine production efficacy and CD107a expression levels of NK-92 cells in three distinct media are evident, with significant disparities observed in cytokine production and CD107a expression in Xuri T-cell media. While the data does not exhibit statistical significance due to the sample size, we have represented the trends using mean fluorescence intensity. This approach allows us to observe the average trend across the total of 10,000 cells analyzed. These findings illustrate that NK cells cultured in Xuri T-cell media are well-suited not only for cell expansion but also for enhancing cell efficacy.

## Evaluation of expansion efficiency of genetically engineered NK-92 cells (CD16-NK-92 or TCR-NK-92)

Next, experiments were conducted to confirm that Xuri T-cell expansion medium can successfully support the expansion of genetically engineered NK-92 cells. CD16-NK-92 cells were used here (purchased from ATCC), which are NK-92 cells expressing Fc receptors. In addition, to assess NK-92 expressing T cell receptors, NK-92 cells transduced by lentivirus expression were prepared as in previous studies. Both cell lines were cultured for 20 days to confirm proliferation efficiency.

The two types of genetically modified cells were cultured in Xuri T-cell expansion medium to check cell concentration, expansion efficiency, viability, and proliferation factors. It was confirmed that both CD16-NK-92 expressing Fc receptor or TCR (**Fig 4A–4D**) were efficiently

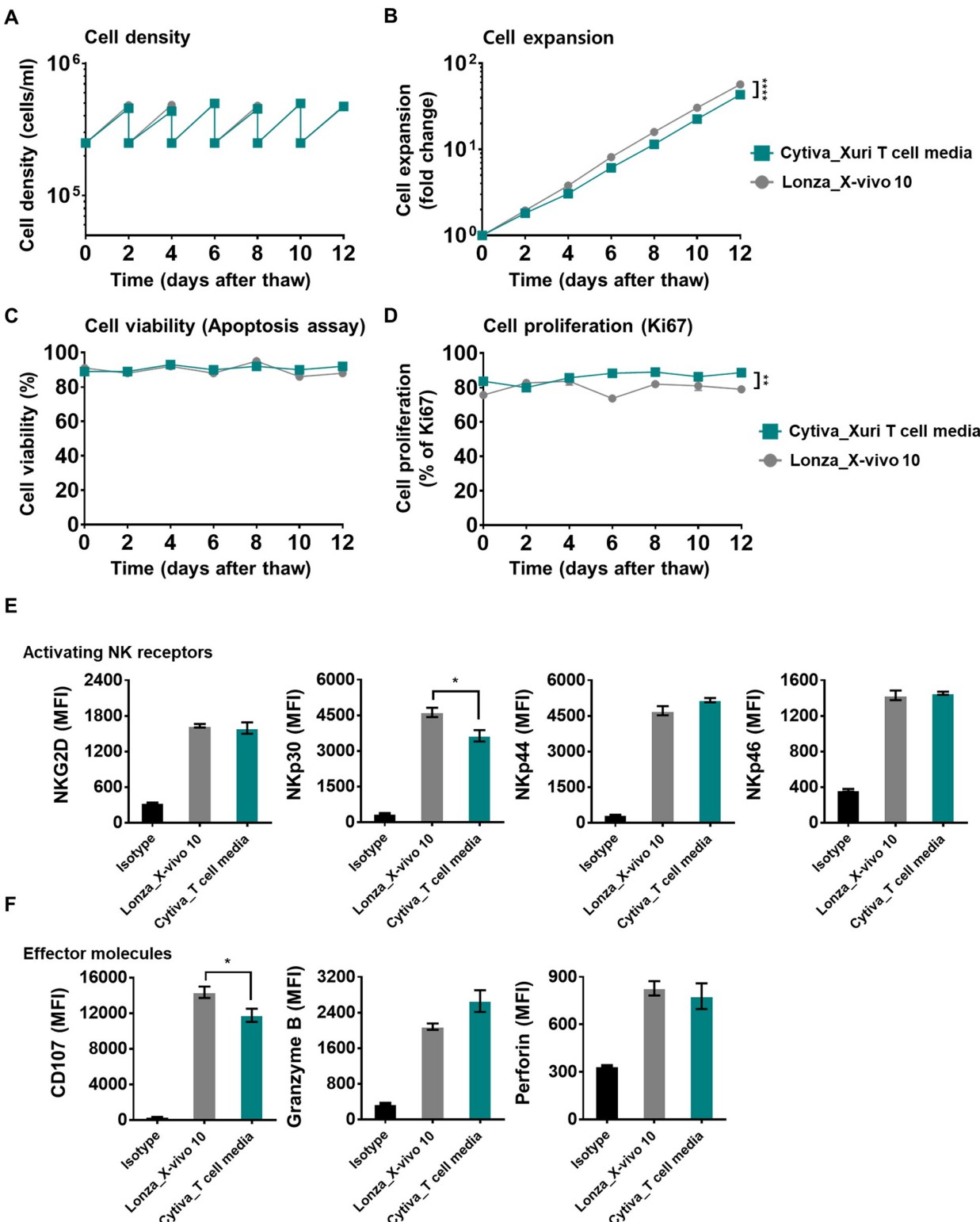

**Fig 3. Assessment of expansion efficiency of NK-92 cells according to the culture medium.** For NK-92 cells, (A) cell concentration, (B) expansion, (C) viability, and (D) proliferation rate were evaluated every two days. NK cells from cultures of two media were analyzed by flow cytometry for expression of various cell-surface receptors. The expression level of cell (E) activator and (F) effector factors was confirmed after culture up to 12 days. To estimate the change in receptor expression, mean fluorescence intensity (MFI) ratios for each receptor were depicted as measured by flow cytometry. Error bars represent standard error of the mean [25]. p value:* $< 0.05$, ** $< 0.01$, **** $< 0.0001$.

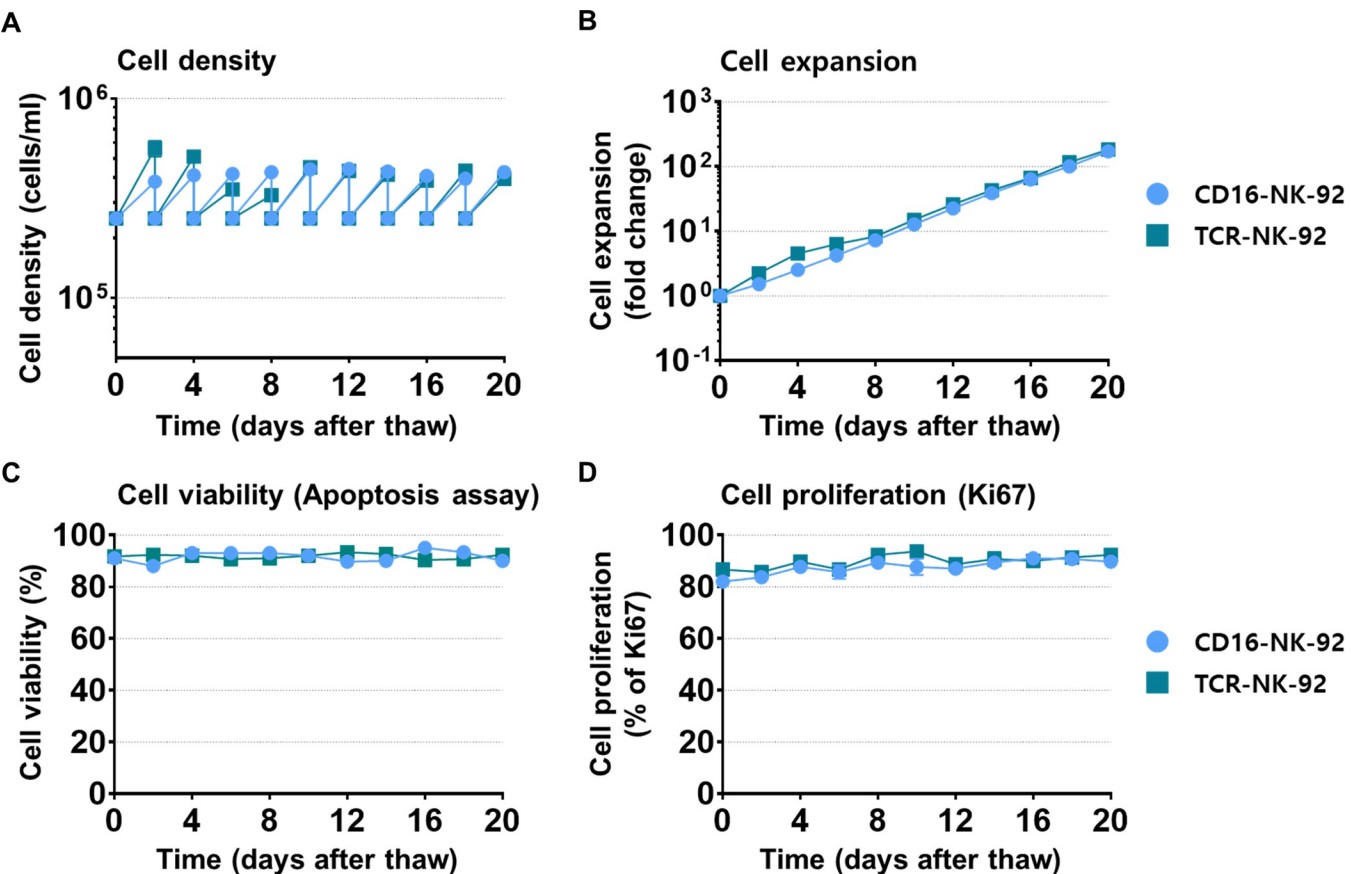

**Fig 4. Assessment of culture efficiency of genetically engineered NK-92 cells.** Genetically engineered NK-92 cells were cultured in Xuri T cell expansion medium to assess (A) cell concentration in culture, (B) expansion efficiency, (C) viability, and (D) proliferation factors. Error bars represent standard error of the mean [25].

expanded in Xuri T-cell expansion medium. CD16-NK-92 cells proliferated 182-fold during the culture period, the viability was 87–95%; additionally, growth factor expression was shown in 82–92% of cells. TCR-expressing NK-92 cells proliferated 205-fold during the culture period, the viability was observed to be 92–95% and growth factor expression was shown in 85–96% of cells. These results suggest that Xuri T-cell expansion medium Xuri T cell expansion medium is effective for both proliferation and phenotypic maintenance of genetically engineered NK-92.

## Expansion efficacy in the Xuri W25 cell expansion system (NK-92 and TCR-NK-92)

The Xuri W25 cell expansion system is a functionally closed system for cell culture wherein the cells grow in a disposable sterile bag within a temperature and air-controlled bioreactor. Based on the results obtained through T flask culture, the appropriate medium choice and culture conditions were set for testing cell proliferation in the Xuri W25 cell expansion system.

The Xuri W25 system, if used with a 2L or a 10L CellBag bioreactor, is designed to operate optimally at a minimal working volume of 300 mL and 500 mL, respectively. Thus, it is not recommended to initiate expansion in lower volumes and/or lower cell doses due to this being out of the recommended operating range of the bioreactor. Therefore, in initial optimization

**Table 1. Xuri W25 setpoints.**

| Parameter | Setpoint |
|---|---|
| pH (pH units) | ~7.2–7.4 |
| Temperature (˚C) | 37˚C |
| Airflow (L/min) | 0.1 |
| Rock rate (RPM) | 6 |
| Rocking angle (˚) | 2 |

experiments, we initiated the cultures in flasks and transferred the cells into the bioreactor at around day 6~9 when sufficient numbers of cells were reached. The bioreactor cultures were started with $1 \times 10^8$ cells/mL (2L scale bag) in 500 mL or $2.5 \times 10^8$ cells/mL in 1000 mL (10L scale bag).

The setpoints for the bioreactor were as shown in **Table 1** below at all timepoints.

After cell inoculation into the Xuri CellBag, the medium was added every two days to maintain the cells at $2.5 \times 10^5$ cells per mL until the maximum working volume of the CellBag was reached (1L and 5L for a 2L CellBag bioreactor and 10L CellBag bioreactor, respectively). Half of the spent medium was removed using a perfusion line when the cell concentration reached $0.5 \sim 1.0 \times 10^6$ cells/mL. After that, cells were maintained at $2.5 \times 10^5$ cells per mL by adding an equivalent volume of medium as a batch feed. After reaching the maximum volume, perfusion was performed (method 1 or 2). The cells were sampled and counted every other day and assessed for cell expansion efficacy and viability.

As shown in **Fig 5A**, culture in the Xuri W25 cell expansion system was carried out in a 2L scale bag and 10L scale bag. The control group, which was cultured in T flasks, required 20 to 21 days to reach 100-fold culture expansion. Compared to the control group, 14 to 15 days was required to reach 100-fold expansion for the 2L scale bag. The culture performed in a 10L scale bag required 10 to 11 days in the first condition (denoted "method 1" in Materials and Methods section) and 8 to 9 days in method 2, (denoted "method 2" in Materials and methods section) to reach 100-fold expansion.

Based on the culture media conditions obtained in previous experiments, the experiment was repeated three times. It was confirmed that all three cultures reached 100-fold within nine days (**Fig 5B**). In addition, it was confirmed that the expression levels of cell surface activator receptors (NKG2D, NKp30, NKp44, NKp46), effector molecules (CD107a, DNAM-1, Granzyme B, Perforin), and inhibitory receptors (KIR2D, TIGIT, TIM3, PD-1) were also kept constant in all batches (**Fig 5C–5E**).

This suggests that the Xuri W25 is suitable as a closed culture system for clinical research and commercial production of NK-92 cells, maintaining both excellent expansion over time as well as retaining the phenotype necessary for NK-92 identity and potency.

## Discussion

Clinical studies using T cells or NK cells are actively being conducted in cell therapy, especially in the field of cancer immunotherapy. Most of the CAR-T cell therapies approved to date use an autologous production model in which T cells are selected from PBMCs isolated from patients, genetically modified, and then proliferated and infused back into the same patients. However, this method has its limitations, such as requiring already sick patients to be further apheresed and hospitalized, as well as the cell therapy production facility being close to the infusion clinic for timely delivery of the therapy to the patient. The time to culture the cells, which must be produced on demand rather than thawed from off-the-shelf stocks, adds to the

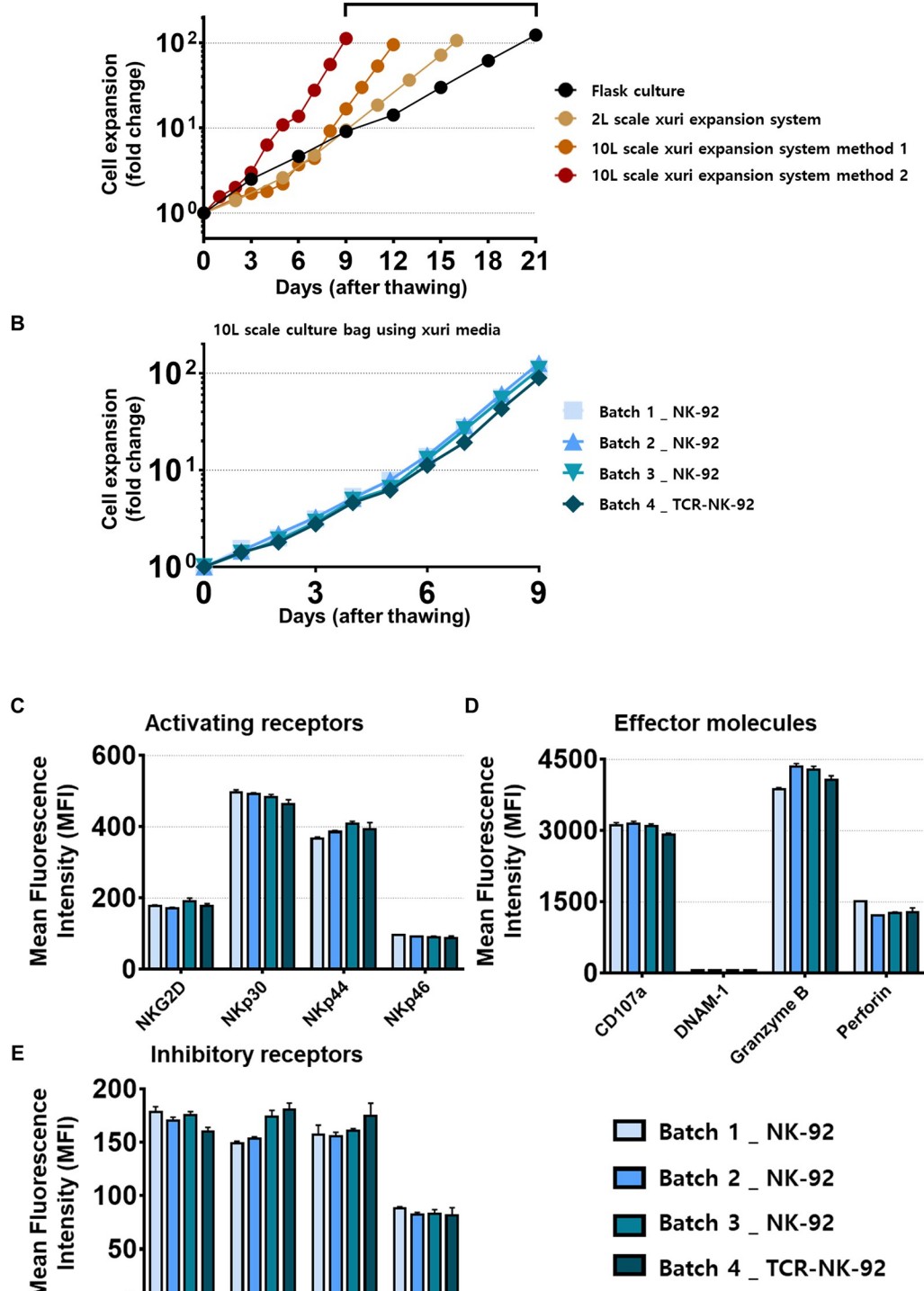

**Fig 5. Expansion efficacy in the Xuri W25 cell expansion system.** Cells were cultured using a Xuri W25 cell expansion system. The bioreactor was used to control parameters like rock rate, angle, temperature- and gas mix in an automated manner. (A) Expansion efficiency according to cell culture bag capacity. (B) Evaluation of reproducibility of expansion efficiency using NK-92 and TCR-NK-92 cells in the Xuri W25 cell expansion system. The expression level of cell (C) activation receptors, (D) effector molecules, and (E) inhibitory receptors was confirmed after culture up to 12 days. To

estimate the change in receptor expression, MFI ratios for each receptor were depicted as measured by flow cytometry. Error bars represent standard error of the mean [25].

time the patient must wait prior to receiving treatment. This study evaluated a cryopreservation production process that can produce 'off-the-shelf' cell therapy products in large scale without such restrictions on the patient and the production facility. In this study, we aimed to evaluate a proof-of-concept for the process of freezing, thawing, and expansion of NK-92 cells, with the eventual application to mass production of cell therapeutics in ready-made form.

Five key cryopreservation parameters were assessed in this study–cooling rate, cell concentration, different cryoprotectants (CPAs), induced ice nucleation, and storage time of the cells [16].

A cooling rate of 1°C/min was found optimal in the NK cell system when assessed 3 days-post thaw. This is typical of similar cell types, further supporting the use of this cooling rate [14, 30, 31]. However, differences were noted between the different cryoprotectants used. Cooling rates must be tightly controlled during cryopreservation to an optimal range. As a system cools, ice forms in the extracellular space locking away water molecules, and only water molecules, as ice. All remaining solutes and cells remain in increasingly concentrated channels between ice crystals as the temperature falls [32, 33]. This increase in solute concentration drives crucial cellular dehydration during the cryopreservation process. If this dehydration does not occur–for example by using a cooling rate which is too fast, and in doing so, not allowing enough time for dehydration, ice may form inside the cell membrane. Such intracellular ice is usual fatal to cells and so cooling too quickly results in lower post-thaw cell viability and recovery. Conversely, as CPAs and concentrated solutes are cytotoxic, cooling too slowly can result in cell damage and death due to prolonged exposure to these high solute concentrations [23, 24].

Cell viability is not affected even when cryopreservation is carried out at a concentration of 10 times the concentration normally used for freezing cells. That is, by minimizing the use of cryopreservation solution, easier removal of CPA after cell thawing can be expected.

Regardless of CPA used (DMSO-containing or DMSO-free), cooling at 0.5°C/min resulted in increasing cell death at increasing timepoints from thaw, possibly indicating severe damage caused by this toxicity. Interestingly, at faster rates of cooling, 2°C/min, CS10 and cryoSOfree has poorer outcome, likely caused by intracellular ice forming during cooling, while the same was not shown with the STEM CELL BANKER solution. This may indicate that NK cells cryopreserved using STEM CELL BANKER allow faster cell dehydration during cooling, perhaps through the use of extra-cellular solutes to increase dehydration rates [19]. Flexibility in allowing more rapid cooling rates is practically useful in the manufacturing process, as using more rapid cooling rates without impacting the post-thaw outcome allows for reduced cryopreservation time, thereby reducing associated costs, or allowing more cryopreservation runs to be performed in a given timeframe.

NK cells were found to be sensitive to ice nucleation in this study, however this only became apparent 72 hours post-thaw, highlighting the need for cell assessment not only immediately on thaw but at later timepoints too when all the impacts of cryopreservation become apparent. Cryovials were nucleated at -7°C, which was compared to no ice nucleation, typically occurring at -10 to -15°C in these volumes [34]. Ice nucleation is perhaps one of the least-controlled parameters during the cryopreservation in the manufacturing process. Ice does not automatically form at a solutions equilibrium melting point, rather the system will remain in a liquid state, known as supercooled, until ice spontaneously forms, either on a particle within the vial, on the wall, or due to a random alignment of atoms. Ice nucleation becomes increasingly likely

at increasingly low temperatures, but due to the stochastic nature of the processes cannot be precisely predicted [34–36]. Current methods to induce ice nucleation are either ineffective (such as plunge steps) [35] or introduce user variability to the cryopreservation process (manual nucleation) [24]. Ice nucleating in very supercooled solutions can damage cells as ice forms more rapidly at lower temperatures and introduces a large temperature discontinuity during cooling. The effect of ice nucleation on NK cells has not previously been reported–in some PBMC populations such as CD45+ cells ice nucleation has been seen to damage cells, but this is not repeated in CD34+ cells [35], with similar heterogeneity seen in PBMCs more generally [15].

For cell concentration–the higher the concentration of cells cryopreserved the fewer cryoprotectants need to be used, which makes washing of transfusion processes simpler, as well as allowing for more practical storage of a large number of therapeutic samples.

To allow economies of scale, cryopreserved cells should remain viable for long periods in storage. This requires storage of the cells below the glass-transition temperature of the CPA solution [37]. Below such a temperature, all biological and chemical processes cease, rendering the cells stable indefinitely so long as temperature is maintained, and cells are protected from physical damage. This temperature has previously been measured at -123˚C in CS10 [37]. STEM CELL BANKER also provided stable storage. CryoSOFree maintained viability at all timepoints, however this was lower in all cases indicating limited cryoprotection during the cryopreservation process, CryoSOFree also failed to maintain proliferation ability within the cells, indicating more fundamental damage to the cells.

Currently, various formulations are being tried as cell cryoprotectants in clinical research for cell therapy, and most of all use DMSO, however there has recently been demand to move away from DMSO due to potential toxic side-effects [38]. The most widely used CPA is CryoStor CS10. CryoStor CS10 is a serum-free, animal-free cryopreservation medium containing 10% dimethyl sulfoxide (DMSO). This has been compared against two CPAs without DMSO; namely, CryoSOfree DMSO free and STEM-CELL BANKER DMSO FREE.

STEM CELL BANKER DMSO FREE shows a high-efficiency of post freeze-thaw cell survival rate DMSO, so it is suggested as a suitable CPA for clinical research.

Assessment of post thaw culture of cells is also key. It has long been known that viability immediately on thaw tends to give artificially high results–viability and function fall post-thaw to a nadir around 24-48h post-thaw, cell type and cryoprocess dependent, due to delayed onset cell death, commonly through apoptotic pathways [37–41]. Therefore, the subsequent longer term culture must be assessed in relation to cryopreservation.

We evaluated the post-thaw proliferation efficiency as a function of culture medium using during cell proliferation. Lonza's X-VIVO 10 is a commonly used culture medium for the proliferation of immune cells such as T or NK in clinical studies of cell therapeutics. Cytiva and Miltenyi are offering Xuri T cell expansion medium and NK MACS products for culturing immune cells or NK cells, and thus represent good options to test for comparability or superiority to X-VIVO 10. We observed a peculiarity in the culture process immediately after the thawing of NK-92 cells. After thawing NK-92 cells in X-VIVO 10, it was confirmed that cell aggregation was observed when the culture was started. To find the root cause of this problem, an experiment of thawing was performed in three culture solutions. Aggregation was observed when NK-92s were thawed in both X-VIVO 10 and in NK MACS. X-VIVO 10 took some time to stabilize until the cells started to proliferate normally. Interestingly, such aggregation was not observed in NK-92 cells thawed in Xuri T cell expansion medium. It was confirmed that the cells were rapidly stabilized after thawing and proliferation started immediately. This result shows that Xuri T cell expansion medium is functionally superior compared to other products in stabilizing cells for proliferation immediately after thawing.

Subsequently, we compared the proliferation efficiency of NK-92 cells in Xuri T cell expansion medium and X-VIVO 10 medium. Although X-VIVO 10 is not adequate for cell stability immediately after thawing, it has excellent proliferation efficiency in the culture process. This proliferation efficiency was reproduced in Xuri T cell expansion medium. Specifically, in the expansion process after cell stabilization, there were no functional differences in cell concentration, proliferation rate, viability, and expression of proliferation factors. Also, it did not affect the expression of cell surface factors or effector molecules after proliferation. These results suggest that there is no functional difference between Xuri T cell expansion medium and X-VIVO 10 medium in terms of cell proliferation.

As in Table 2 above (see Methods), we tested different modifications of the NK92 line across different studies. This allowed us to assess the impact of expansion and cryopreservation on various modifications of the cell line. An advantage of this approach is that it enabled a broader and more representative data set the ways in which NK92 cells may be modified and used in the field clinically. A disadvantage of this approach is that it renders a head-to-head comparison of outcomes difficult across the different studies, since each line was modified with a slightly different receptor. That said, all results within a single experiment were internally consistent and used a single NK92 line.

As discussed previously, clinical studies using genetically engineered immune cells are actively underway. Therefore, whether there is a difference in cell expansion after genetic modification is also an essential factor to test. We evaluated the cell proliferation efficiency of genetically modified NK-92 cells cultured in Xuri T cell expansion medium. The proliferation efficiency was evaluated using CD16-NK-92 cells (commercially available as a modified variant of NK-92 with Fc receptor expression) and TCR-NK-92 cells, transduced with lentivirus in house with T cell receptor complex expression. During expansion of genetically engineered NK-92 cells, there were no functional differences in cell concentration, proliferation rate, viability, and expression of proliferation factors. These results suggest no expansion efficacy loss between wild-type NK-92 and genetically engineered NK-92, using Xuri T cell expansion medium, making it suitable for use with modified NK cell therapies.

For 'off-the-shelf' production, large-scale capacity is required in one batch production. In particular, for clinical research, the production process should ideally be carried out in functionally closed, automated, large scale expansion bioreactor. We used Cytiva's Xuri W25 cell expansion system and showed that it was suitable for NK-92 culture among various types of cell expansion equipment. The Xuri W25 cell expansion system allows proliferation to be carried out in a functionally closed cell culture bag, and all processes are digitally controlled via instrument software. It was shown to be suitable for the expansion of cell therapeutics based on NK-92s, and amenable for either clinical research, or for commercial manufacturing. The inclusion of temperature and perfusion control, as well as gentle rocking motion, make it ideally suited for expansion of shear-sensitive cells like immune cells, and the functionally closed

**Table 2. Cells type used in the study.**

| Figure No | Using Cell Type |
| --- | --- |
| Fig 1 | CD3-NK92 |
| Fig 2 | CD3-NK92 |
| Fig 3 | NK92 |
| Fig 4 | CD16-NK92 and TCR-NK92 |
| Fig 5 | NK92 and TCR-NK92 |
| S1 Fig | CD3-NK92 |
| S2 Fig | CD3-NK92 |

system combined with automation and GMP compliance, ensure that it is safe from contamination, and that all data is recorded.

Typically, static culture methods involve the use of either a static flask or a gas permeable cell culture bag. These methods can have high proliferation efficiency with cell lines if the culture is batch fed or split often to maintain cells in a continuous state of proliferation, but it is labor-intensive and difficult to continually change the media in these static systems; moreover, continuous culture expansion eventually affects the growth and properties of cells, and the high degree of open and manual manipulation introduces other risks to the culture, such as contamination. In particular, since it is a static culture, colony formation of NK-92 cells is excessively induced, and the proliferation efficiency decreases as the cell concentration increases.

On the other hand, the Xuri W25 cell expansion system is a continuous culture system, and due to the automated perfusion system that can exchange culture media in real-time, a high proliferation efficiency is maintained throughout the culture period.

We evaluated the efficacy of culture in the Xuri W25 cell expansion system based on various conditions previously verified through flask culture. To recapitulate conditions typically of a small-scale allogeneic NK cell production batch, experiments were performed in both the 2L and 10L scale culture bags. The expansion period of NK-92 cells was shortened by a factor of 1/2.33 (or approximately 42% shorter duration) compared to flask culture in the optimized 10L scale culture condition. To confirm the reproducibility of the culture under this condition, repeated experiments were performed.

Three replicates were performed using NK92 cells, and it was confirmed that reproducibility was maintained in each condition. TCR-NK-92 cells were also cultured under the same conditions, and the same proliferation efficacy was confirmed. There were no differences in expression of the activation receptors, effector molecules, and inhibitory receptors of NK-92 cells across the three replicates. These results show that expansion in a closed system bioreactor enhanced the proliferation efficiency of NK-92 cells and maintained the phenotype of the cells in a reproducible manner.

In summation, we carried out studies to optimize conditions for cell freezing, thawing, and expansion using various NK-92 cell lines and derived meaningful results. Cell therapeutics, particularly those based on NK and NK-92 cells, are a promising solution in the development of novel anti-tumor therapeutics. Optimizing the production process of NK-92 cells is a substantial step toward the development of 'off-the-shelf' cell therapy products. Although the results of this study are positive, further optimization is needed to improve fold-expansion during culture, reduce culture time and demonstrate larger production scales suitable for off-the-shelf production. These studies represent an essential step toward closing and automating the NK expansion workflow, while introducing GMP-available reagents, kits, and hardware that are suitable for NK cell culture. Future directions will look at further automating and scaling up the process.

## Supporting information

**S1 Fig. Evaluation of cell aggregation according to culture media after cell thawing.** NK-92 cells were frozen using seven types of cryoprotective agents. After thawing the frozen cells, (A) Cultured in Xuri T-cell media after 24 hours. (B) Cultured in X-vivo 10 media to observe cell aggregation after 24 hours.
(TIF)

**S2 Fig. Assessment of cytokine production efficacy of NK-92 cells according to the culture medium.** (A) NK-92 cells cultured in three distinct media were subjected to flow cytometry

analysis to evaluate the expression of effector molecules. To quantify the alterations in expression levels, we represented the MFI ratios for each individual molecule as determined through flow cytometry measurements.
(TIF)

## Acknowledgments

This work was supported by the Research Foundation of Yonsei University (no. 6-2017-0104) and the Basic Science Research Program of the National Research Foundation of Korea (NRF) funded by the Ministry of Education, Science, and Technology (2019R1A2C4069993), and supported by Cytiva for providing process development.

## Author Contributions

**Conceptualization:** Youngseon Byeon.

**Data curation:** Seul Lee, Yunjoo Joo, Eun Ji Lee, Youngseon Byeon, Jung-Yub Lee, Jeeheon Kang, Byoung Chul Cho.

**Formal analysis:** Youngseon Byeon, Byoung Chul Cho.

**Funding acquisition:** Byoung Chul Cho.

**Investigation:** Eun Ji Lee, Youngseon Byeon, Byoung Chul Cho.

**Methodology:** Youngseon Byeon, Peter Kilbride, Rohin K. Iyer, Mandy C. French.

**Project administration:** Jae-Hwan Kim, Kyoung-Ho Pyo, Young Seob Kim, Sun Min Lim, Peter Kilbride, Rohin K. Iyer, Hyesin Byun, Byoung Chul Cho.

**Supervision:** Byoung Chul Cho.

**Visualization:** Youngseon Byeon, Mingming Li, Mandy C. French, Jung-Yub Lee, Jeeheon Kang.

**Writing – original draft:** Seul Lee, Youngseon Byeon, Peter Kilbride, Byoung Chul Cho.

**Writing – review & editing:** Byoung Chul Cho.

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
