## [Decision Letter · Decision Letter 0]

10 Aug 2023

PONE-D-23-19842Successful expansion and cryopreservation of human natural killer cell line NK-92 for clinical manufacturingPLOS ONE

Dear Dr. Cho,

Thank you for submitting your manuscript to PLOS ONE. After careful consideration, we feel that it has merit but does not fully meet PLOS ONE’s publication criteria as it currently stands. Therefore, we invite you to submit a revised version of the manuscript that addresses the points raised during the review process. Please submit your revised manuscript by Sep 24 2023 11:59PM. If you will need more time than this to complete your revisions, please reply to this message or contact the journal office at plosone@plos.org. Please include the following items when submitting your revised manuscript:A rebuttal letter that responds to each point raised by the academic editor and reviewer(s). You should upload this letter as a separate file labeled 'Response to Reviewers'.A marked-up copy of your manuscript that highlights changes made to the original version. You should upload this as a separate file labeled 'Revised Manuscript with Track Changes'.An unmarked version of your revised paper without tracked changes. You should upload this as a separate file labeled 'Manuscript'.If applicable, we recommend that you deposit your laboratory protocols in protocols.io to enhance the reproducibility of your results. Protocols.io assigns your protocol its own identifier (DOI) so that it can be cited independently in the future. For instructions see: https://journals.plos.org/plosone/s/submission-guidelines#loc-laboratory-protocols. Additionally, PLOS ONE offers an option for publishing peer-reviewed Lab Protocol articles, which describe protocols hosted on protocols.io. Read more information on sharing protocols at https://plos.org/protocols?utm_medium=editorial-email&utm_source=authorletters&utm_campaign=protocols.

We look forward to receiving your revised manuscript.

Kind regards,

Syed M. Faisal, Ph.D.

Academic Editor

PLOS ONE

Journal Requirements:

   "NO"

5. We notice that your supplementary figures are included in the manuscript file. Please remove them and upload them with the file type 'Supporting Information'. Please ensure that each Supporting Information file has a legend listed in the manuscript after the references list.

Additional Editor Comments:

Address all queries posed by the learned reviewer where feasible. If the authors believe that certain aspects have not been appropriately addressed, they are required to provide justifications.

Reviewers' comments:

Reviewer's Responses to Questions

**Comments to the Author**

1. Is the manuscript technically sound, and do the data support the conclusions?

Reviewer #1: Yes

Reviewer #2: Yes

Reviewer #3: Yes

Reviewer #4: Yes

2. Has the statistical analysis been performed appropriately and rigorously? 

Reviewer #1: Yes

Reviewer #2: Yes

Reviewer #3: Yes

Reviewer #4: Yes

3. Have the authors made all data underlying the findings in their manuscript fully available?

Reviewer #1: Yes

Reviewer #2: Yes

Reviewer #3: Yes

Reviewer #4: Yes

4. Is the manuscript presented in an intelligible fashion and written in standard English?

Reviewer #1: Yes

Reviewer #2: Yes

Reviewer #3: Yes

Reviewer #4: Yes

5. Review Comments to the Author

Reviewer #1: The manuscript by Seul Lee et al., titled "Successful expansion and cryopreservation of human natural killer cell line NK-92 for clinical manufacturing," has been well written as well as studied manuscript which focuses on the development of a model process for the manufacturing of natural killer (NK) cells, which have shown promise in treating hematologic cancer. In this study, the researchers utilized Cytiva hardware and media to expand the NK92 cell line, aiming to establish a GMP (Good Manufacturing Practice) and clinical manufacturing process for NK cells. By automating and connecting unit operations within the workflow, they achieved improved expansion and cryopreservation of NK92 cells. The cellular outputs and performance were found to be suitable for clinical dosing regimens, serving as a proof-of-concept for future clinical and commercial manufacturing of NK cells.

Advantage of this study:

In summary, this research demonstrates the development of an optimized manufacturing process for NK cells, addressing the current limitations of manual and outdated workflows. The use of advanced technologies and media resulted in superior expansion and cryopreservation outcomes for NK92 cells, paving the way for potential clinical and commercial applications of NK cell therapies. The successful scale-up of NK cell production requires optimized expansion media and cryopreservation conditions to ensure high cell viability and recovery after thawing.

Limitation of this study: While the presented study provides valuable insights into the cryopreservation of NK-92 cells, there are some weaknesses that should be acknowledged:

1. The study focuses specifically on the cryopreservation of NK-92 cells and does not explore the comparison with other types of cells or cell lines. The findings and conclusions may not necessarily generalize to other cell types, and further research is needed to validate the effectiveness of the proposed method across different cell lines.

2. The study does not directly compare the proposed cryopreservation method with conventional cryopreservation techniques or existing protocols. A comparative analysis would have provided a better understanding of the advantages and limitations of the proposed method in relation to established approaches.

3.The study primarily evaluates cell viability and proliferation after thawing, with a focus on survival rates. However, it lacks a comprehensive assessment of other functional characteristics, such as cell functionality, phenotype, genetic stability, and immune response. Assessing these factors would provide a more comprehensive understanding of the impact of the cryopreservation method on NK-92 cell functionality and therapeutic potential.

4. Although the study examines the viability and proliferation of NK-92 cells after thawing, it does not provide long-term data on cell functionality or stability over extended periods. Assessing the long-term effects of cryopreservation on NK-92 cells, such as their cytotoxic activity, cytokine production, and immune modulation, would be valuable in determining the overall suitability of the method for clinical applications.

5. While the study discusses the potential clinical applications of NK-92 cells, it does not include any in vivo or clinical data. Further research involving animal models or clinical trials is necessary to evaluate the effectiveness and safety of the proposed cryopreservation method in a clinical setting.

Minor comment:

A minor comment to improve the study would be to include an in vitro experiment to validate the efficacy of the NK-92 cell line after thawing, specifically focusing on immunomodulation. This would provide valuable insights into the functional capabilities of the cryopreserved NK-92 cells in a physiological setting and further validate their therapeutic potential. Conducting an in vitro experiment would allow for a more comprehensive assessment of the immunomodulatory effects, including tumor targeting via cytokine such as (TNF-α) production capacity and compare it with other cell line cultured in different media. This additional data would strengthen the study's findings and provide more robust evidence for the clinical applicability of the cryopreservation method for NK-92 cells.

Reviewer #2: The review article entitled “Successful expansion and cryopreservation of human 2 natural killer cell line NK-92 for clinical manufacturing” is a nice discovery for the cell line development for cancer research. The study proposes an effective method for the expansion and cryopreservation of NK-92 cells for clinical research and industrialization, which is going to be helpful for future impact in cancer research. In addition to commercially available cryopreservation method, author developed a new culture kinetics of Genetically modified NK-92 Cells. For them I have a quick question:

1. The author mentioned in the Fig-1 (Culture using manual method in T flask): the NK92 cells were thawed in culture media supplemented with 5% heat-inactivated human AB serum 250 U/mL of Proleukin (Novartis) and splitted on 3or 4th day to avoid cell-cell contact. My question is NK cells are already fragile and they need more no cells and at least 10% of FBS for grow with supplementing media and proliferate. I have a doubt how they grew on 5% FBS and ready to split on 3rd day.

2. Didn’t the cell clumping affect the cell viability during cryopreservation and revival of NK-92 cells in all 3 methods. What was the optimum time to revive NK-92 cells in all 3 conditions as you mention Vs normal preservation.

3. How did you optimize that cryo-bags are more important as compared to cryovials?

4. Suggestion for Fig-2, can you please provide a good quality picture for publication. Some figures are looking so blurry and please try to take a picture by avoiding debris.

5. The authors check the cell receptor for activating /effector/ inhibitory receptor on the cells, so my question is does the author cocultured NK cells with some tumor cells and check the expression levels of those molecule after coculturing and tumor killing assay. I think you should include this experiment in this publication.

6. Have you ever checked, does this NK92 cells modify the T cell activation specially CD8+ T cells and chemokine/cytokine receptor modulation. Please elaborate this section.

7. Have you ever got chance to mimic this kind of NK cell production in mice and did the evaluation for mechanistic part for antitumor immunity (adaptive immunity generation).

The writing style and the clarity of the exposition are fine, conclusions are clear. The abstract section needs to be organized in scientific language and must present the theme of the manuscript significantly. Additionally, the conventional cryopreservation mechanisms of immune cells specially NK cells need more attention and author did a good job, amazing work for NK92 cell stability after thawing, evaluating the cell expansion efficiency of NK92 and genetically engineered NK-92 cells. This manuscript provides a topic of interest to the researchers in this field and has a potential for possible publication in this journal after some minor corrections.

• Figure should be formatted as per the journal’s guidelines. Please make one format for all the figures space.

• Scientific names and terminology like in vitro/in vivo should be italicized throughout the manuscript.

• All the abbreviations should be clearly stated throughout the manuscript.

• Several spelling and grammar errors need to be corrected.

• The resolution of figures should be as per the standard format.

• After careful proofreading and incorporation of suggested changes, the manuscript can be considered for publication.

Reviewer #3: Lee et al., have worked on optimisation of cryopreservation and expansion conditions for NK-92 cells as well as TCR-NK-92 cells.Overall, the authors should be commended for their design of study is so systematic. They have optimised conditions for cryopreservation of the NK-92 cells by comparing the rate of cooling, induced nucleation, different CPAs with different compositions, for an extended period of time. Also they have compared three GMP grade media formulations for optimum expansion of NK-92 cells and demonstrated a method to scale up the expansion process using Cytiva hardware and media and shown how the continuous closed culture method can improvise the current expansion method of NK-92 cells as well as TCR-NK-92 cells.

Overall, the manuscript is well written, the experiments are sound.However, there are a few shortcomings that need to be addressed before this manuscript can be considered acceptable for publication.

Major issues

Have the authors perform some cytotoxicity activity for the NK-92 cells post cryopreservation and expansion according to the conditions optimised in this manuscript ? Since the ultimate purpose is the clinical use, and the functional aspects of NK-92 cells remains to be a major concern after cryopreservation. It will be good if this experiment is performed to validate that cytotoxic activity of NK92 cells is not affected in the conditions optimised in this manuscript.

Minor issues

1.Would it not be good to combine the supplementary Figure 1 with Figure 1 (to avoid duplication of data that is already shown in Figure 1).

2.In Figure 2, the X axis title, would it not be more meaningful if we write “days after thawing” rather than “thawing after days”

3.In Supplementary Figure 2 legends, the authors mentioned use of “seven” CPAs but the figure shows only Four.

4.In Figure 5 D, are the cells used in this study DNAM-1 negative?

5.In line 140, what does it mean to acclimate the cells upto 12days since some experiments were also performed at 0 hour.

6. Can the authors make some comments on possibility of number of freeze-thaw cycles with the CPAs used in the study.

The authors have worked on a relevant aspect of optimising cryopreservation and expansion conditions that can help bring the “off the shelf therapeutics” NK-92 cells to clinical regimen with better scalability. After suggested changes, overall the data presented is suitable for publication.

Reviewer #4: Research studies based on designing and developing therapeutic approaches to manage tumors is currently need of the hour due to increase in number of cancer patients worldwide. NK cells therapy has shown promising results in the treatment of hematological cancer symptoms, but the issues associated with it’s large-scale production are yet to be effectively addressed . The present manuscript is authors’ successful effort to deal with this wherein they have designed an effective method targeting the expansion and cryopreservation of both NK-92 and genetically modified NK-92 cells for clinical research and pharmaceutical industries. The manuscript is well written and the experiments are skillfully designed and meticulously executed.

I have only a few concerns about the manuscript.

Discussion seems a little lengthy; could be shortened and be written in a more interesting way.

Abbreviations should be defined where used for the first time and then consistent in the manuscript.

6. PLOS authors have the option to publish the peer review history of their article (what does this mean?). If published, this will include your full peer review and any attached files.

Reviewer #1: **Yes: **ZEESHAN AHMAD

Reviewer #2: **Yes: **NIDA MUBIN

Reviewer #3: No

Reviewer #4: **Yes: **Nazim Husain

---

## [Author Response · Author response to Decision Letter 0]

31 Oct 2023

I sincerely appreciate your thoughtful comments. We have made every effort to address your comments and kindly request that you refer to our cover letter for further details.

---

## [Editor Report · Decision Letter 1]

9 Nov 2023

Successful expansion and cryopreservation of human natural killer cell line NK-92 for clinical manufacturing

PONE-D-23-19842R1

Dear Dr. Cho,

We’re pleased to inform you that your manuscript has been judged scientifically suitable for publication and will be formally accepted for publication once it meets all outstanding technical requirements.

Kind regards,

Syed M. Faisal, Ph.D.

Academic Editor

PLOS ONE

**Additional Editor Comments:**

I would like to bring to your attention a matter that requires your prompt attention to ensure the integrity and clarity of the published work. It has been noted that in **Figure 2C and 2D**, as well as in **Supplementary Figure 2**, there are missing statistical error bars on the graphs.

I kindly ask that you address this oversight by adding the appropriate error bars to these figures during the production office corrections and ensure that the statistical details corresponding to these error bars (such as standard deviation or standard error of the mean) are clearly indicated in the figure legends. 
---

## [Editor Report · Acceptance letter]

2 Feb 2024

PONE-D-23-19842R1 

PLOS ONE

Dear Dr. Cho, 

I'm pleased to inform you that your manuscript has been deemed suitable for publication in PLOS ONE. Congratulations! Your manuscript is now being handed over to our production team.

Kind regards, 

on behalf of

Dr. Syed M. Faisal 

Academic Editor

PLOS ONE